# Overview of Hybrid Energy Storage Systems Combined with RES in Poland

**Piotr Hylla [1],\* , Tomasz Trawiński [2] , Bartosz Polnik [1], Wojciech Burlikowski [3] and Dariusz Prostański [1]**

[1]  KOMAG Institute of Mining Technology, 44-100 Gliwice, Poland; bpolnik@komag.eu (B.P.); dprostanski@komag.eu (D.P.)
[2]  Faculty of Electrical Engineering, Silesian University of Technology, 44-100 Gliwice, Poland; tomasz.trawinski@polsl.pl
[3]  Department of Mechatronics, Silesian University of Technology, 44-100 Gliwice, Poland; wojciech.burlikowski@polsl.pl
\*  Correspondence: phylla@komag.eu

**Abstract:** This article reviews the most popular energy storage technologies and hybrid energy storage systems. With the dynamic development of the sector of renewable energy sources, it has become necessary to design and implement solutions that enable the maximum use of the energy obtained; for this purpose, an energy storage device is suggested. The most popular methods of electric energy storage are described, with an indication of the features of each technology, along with the presentation of the advantages and disadvantages of a given storage reservoir. Next, hybrid energy storage systems are presented along with their suggested applications and advantages resulting from the hybridization of technologically diverse energy storage systems.

**Keywords:** RES; HESS; ESS; BESS; FESS

## 1. Introduction

Electric energy in Poland is increasingly produced with the use of environmentally friendly renewable energy sources (RES) [1]. According to the data published by the Energy Market Agency, the dynamics of the increase in the number of installed RES sources in November 2022 was 135.2% compared with the previous year. So-called green energy [2] is produced using wind turbines [3], hydro power plants [4], biogas plants [5], photovoltaics [6] and biomass power plants [7]. Photovoltaics (PV) is currently the most popular RES technology in Poland, covering installations with a total capacity of 12,000 MW. Photovoltaic (PV) installations are also the fastest growing sector of renewable energy sources, with an increase in the amount of installed capacity at a level of 167% (November 2022) compared with the previous year. The second largest RES source in terms of installed capacity, reaching nearly 7900 MW (November 2022), are wind turbines, with a growth rate of 112.3% compared with the previous year. The renewable energy sources are environmentally friendly but are also unstable and unpredictable [8].

The purpose of this article is to present the possibility of increasing the share of renewable energy sources in the total energy production through the use of electricity storage facilities. Due to the lack of sufficient infrastructure to balance energy from RES, it is not possible to use renewable energy sources without certain restrictions [9]. Further, the arguments will be presented to confirm the following theses:

- By using state-of-the-art electricity storage installations, it is possible to increase the flexibility of operation of conventional and nuclear power plants, stabilizing their cooperation with unstable sources, e.g., RES [10,11].
- Limitations resulting from the storage technologies, including electric, mechanical, chemical and thermal storage tanks characterized by different parameters and natures of operation, hinder the maximum use of electric energy from RES [12].

- Hybridization of energy storage will enable the adjustment of energy storage parameters to the recipient needs. By analyzing the dynamic growth of RES and the gradual decarbonization of the power industry, the demand for short-, medium- and long-term energy storage is noticed [13].
- When using different sources and energy storage facilities within one electrical installation, it becomes necessary to use systems that integrate the components into a hybrid energy storage system (HESS) [14].

## 2. Materials and Methods

During the analyses of the current state of knowledge, issues related to the use of hybrid energy storage in systems with renewable energy sources were extensively tested. During the literature review, attention was paid to two basic aspects: electricity storage technologies and hybrid energy storage systems. Key research work areas were identified, along with the sources of information and keywords used to search for information. Research work on the current state of knowledge was based on technical and scientific sources such as Google Scholar, IEEE Explorer, Scopus, Science Direct, Web of Science, etc. Information was searched using the following keywords: energy storage systems (ESS), hybrid energy storage systems (HESS), microgrids, renewable energy sources (RES), battery energy storage (BESS), etc. Table 1 shows the number of responses to each keyword in each technical source over the last 5 years.

**Table 1.** Number of responses by keywords and technical sources.

| Keywords | Technical Source | | | | |
|---|---|---|---|---|---|
| | **IEEE Explorer** | **Web of Science** | **Google Scholar** | **Scopus** | **Science Direct** |
| Energy storage system | 24,536 | 83,751 | 163,000 | 80,891 | 313,522 |
| Hybrid energy storage system | 4252 | 12,649 | 145,000 | 12,168 | 98,408 |
| Microgrids | 15,103 | 12,776 | 17,500 | 18,852 | 12,293 |
| Renewable energy sources | 25,732 | 45,417 | 115,000 | 51,493 | 174,884 |
| Battery energy storage system | 10,572 | 27,208 | 42,600 | 28,066 | 86,747 |

This paper analyzes the current state of knowledge in the fields of:

- Electricity storage technology;
- Used or proposed solutions of hybrid electric energy storage.

## 3. Energy Storage Technologies

Electric energy is currently the most universal form of energy transfer. Currently, more and more sectors of the economy are replacing existing solutions with devices powered by electricity. Examples of such industries include the automotive industry (undergoing transformation from combustion drive to electric drive) or the construction sector that increasingly uses high-efficiency heat sources such as heat pumps, thus replacing conventional solid fuel boilers (coal, gas or fuel oil). Contemporary development of the world's economies and the growing population of the world result in an increasing demand for electricity, generated largely by power plants powered by solid fuel that increase $CO_2$ emissions into the atmosphere. In order to reduce the negative effects of the production of electricity from the combustion of solid fuels, photovoltaic panels or wind turbines are used as renewable energy sources. However, as mentioned before, these types of energy sources are unstable and unpredictable in operation. For this reason, the possibilities of using RES as the primary energy source are severely limited because the characteristics of energy processing and its production from RES are shifted in time [15]. Another reason is the increasing number of prosumer photovoltaic installations that negatively affect the parameters of the power grid by increasing the grid voltage. Therefore, it will be important to develop mechanisms that enable balancing energy from RES in the smallest possible area by modernizing the current transmission networks and introducing the systems that

increase self-consumption. A solution to this problem may be energy storage, also known as energy reservoirs. The use of energy storage connected to the low-voltage grid additionally enables the following:

- Creation of microgrids ensuring energy balancing between each energy source and consumers, thus enabling total or partial energy independence [16];
- Security of energy supply for consumers/buildings requiring uninterrupted access to electricity, e.g., hospitals, cold stores, laboratories [16];
- Ability to load the grid with power higher than the connection power without the risk of grid overload or failure. An example of its application are fast charging points for electric vehicles connected to a low-voltage network [17];
- Participation in the complete restoration of electricity supplies after an extensive failure (the so-called Black Start) [16];
- Using energy price differences in given hours by loading the energy storage in an economically favorable period and discharging at the time of increased energy prices [16];
- Improving the quality of the grid's parameters by adjusting the frequency, voltage and reactive power compensation [18].

The first research work on cells enabling energy storage dates back to the beginning of the 19th century, while the first energy storage using potential energy resulting from the difference in water reservoir levels was launched at the beginning of the 20th century. The dynamic development of energy storage can be observed at the end of the 20th century due to the growing demand for electromobility and the development of mobile electronic devices, e.g., mobile phones or laptops. Currently, there are many different technologies that enable energy storage in various forms. The basic division is shown in Figure 1.

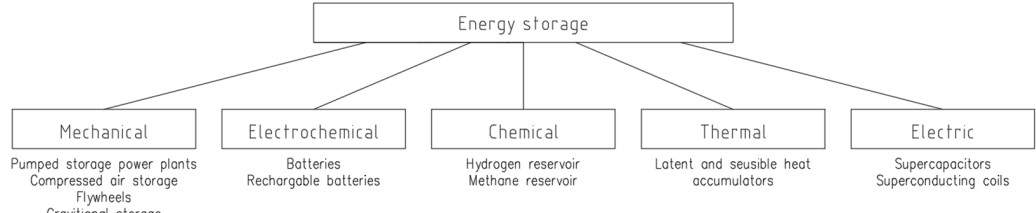

**Figure 1.** Division of energy storage by the technology used.

### 3.1. Pumped Storage Plant

Pumped storage power plants are one of the energy storage methods that use water as the energy storage medium. One of the oldest pumped storage power plants was launched in 1899 in Switzerland on the Sitter River. The water potential energy plant in Sihltal was originally designed to store and generate mechanical energy to power the local textile industry [19]. The principle of operation of pumped storage power plants is based on the conversion of electrical energy into mechanical energy in the form of potential energy of water, enabling energy storage and management. The basic components and the principle of operation are presented in Figure 2. Pumped storage power plants usually consist of two reservoirs, upper and lower. When the network load is low, water from the lower reservoir is pumped to the upper reservoir and, in the case of increased electricity consumption, water from the upper tank is released and (flowing through the generator) is brought to the lower tank to generate electricity. Thus, a pumped storage power plant enables the transformation of surplus energy in the network resulting from reduced energy consumption into potential energy of water, which is converted back into electricity at the peak load of the power grid. In addition, facilities of this type are used for frequency modulation, phase modulation and voltage stabilization in power systems [20]. Electric energy stored in pumped storage tanks can be calculated using Formula (1).

$$E_{el} = \eta E_p \tag{1}$$

where $\eta$ is total efficiency and $E_p$ is the potential energy of water according to the following Formula (2):

$$E_p = mgh = \rho V g h \tag{2}$$

where $m$ is the mass of water in the upper reservoir, $\rho$ is the density of stored liquid, $V$ is the water volume in the upper reservoir, $g$ is the standard gravity and $h$ is the difference in height between the water surfaces in both reservoirs.

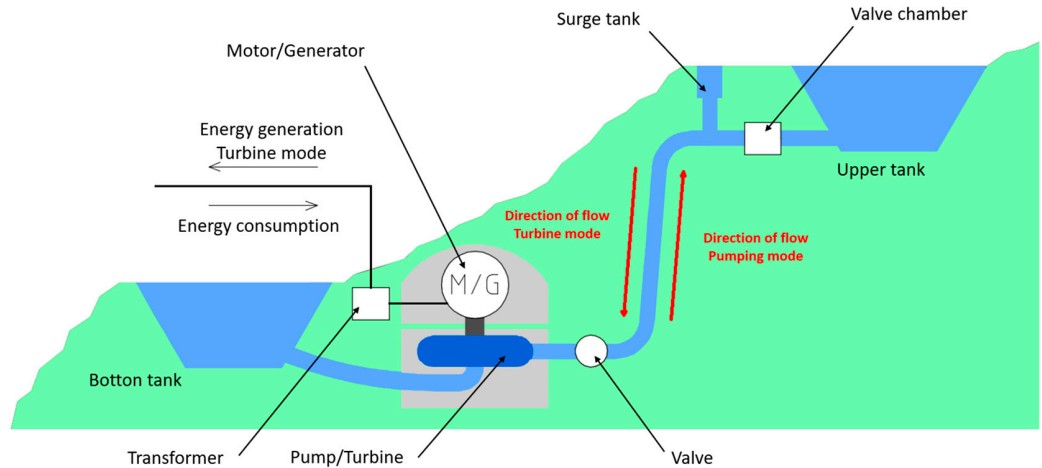

**Figure 2.** Principle of operation of pumped storage power plant [21].

Pumped storage power plants have been the most common method of energy storage for several years, with as much as 95% of the resources with a capacity of 184 GW. It is currently estimated that this technology will continue to be developed and used in the future [22]. Table 2 presents the 10 largest pumped storage power plants in the world [22–31], while Table 3 presents the 5 largest pumped storage power plants currently operating in Poland. Table 4 shows the main advantages and disadvantages of pumped storage power plants.

**Table 2.** Largest pumped storage power plants in the world.

|  | Name of the Object | Country | Max. Power (MW) |
|---|---|---|---|
| 1 | Fengning Pumped Storage Power Station [23] | China | 3600 |
| 2 | Bath County Pumped Storage Station [24] | USA | 3003 |
| 3 | Huizhou Pumped Storage Power Station [25] | China | 2448 |
| 4 | Guangdong Pumped Storage Power Station [26] | China | 2400 |
| 5 | Meizhou Pumped Storage Power Station [27] | China | 2400 |
| 6 | Changlongshan Pumped Storage Power Station [28] | China | 2100 |
| 7 | Okutataragi Pumped Storage Power Station [29] | Japan | 1932 |
| 8 | Ludington Pumped Storage Power Plant [30] | USA | 1872 |
| 9 | Tianhuangping Pumped Storage Power Station [31] | China | 1836 |
| 10 | Tumut-3 [32] | Australia | 1800 |

**Table 3.** Largest pumped storage power plants in Poland.

|  | Name of the Object | Max. Power (MW) | Storage Capacity (GWh) |
|---|---|---|---|
| 1 | Elektrownia Żarnowiec [33] | 716–800 | 3.6 |
| 2 | Elektrownia Porąbka-Żar [34] | 500–540 | 2 |
| 3 | Zespół Elektrowni Wodnych Solina-Myczkowce [35] | 199 | 1.3 |
| 4 | Elektrownia Żydowo [36] | 167 | 0.7 |
| 5 | Elektrownia Czorsztyn-Niedzica-Sromowce Wyżne [35] | 94.6 | 1 G |

**Table 4.** Advantages and disadvantages of pumped storage power plants [19–23].

| Advantages | Disadvantages |
|---|---|
| <ul><li>Possibility of long-term energy storage;</li><li>Possibility of smooth transition from charging to discharging mode;</li><li>Efficiency at levels of 65–85%;</li><li>No influence of the number of charge/discharge cycles on the energy storage capacity;</li><li>It takes a few minutes to reach full power;</li><li>Used to control fluctuations in the load of power grids;</li><li>Long life of the energy storage compared with other technologies.</li></ul> | <ul><li>High investment costs;</li><li>The need to interfere in the natural environment when creating reservoirs;</li><li>Possibility of disturbing the natural flora and fauna in the reservoirs;</li><li>Impact of weather conditions, especially periods of drought, on the operation of the installation;</li><li>Long construction time of pumped storage power plants;</li><li>Long payback time;</li><li>Strong impact of geographical conditions for the construction of the upper and lower reservoir.</li></ul> |

### *3.2. Kinetic Energy Storage*

Kinetic energy storage (FESS (flywheel storage system)) is another energy storage technology. FESS uses the kinetic energy of a rotating object (flywheel) to store energy [37]. One of the first applications of flywheels was the potter's wheel that used this effect to hold energy under its own inertia. Over time, flywheels found application in simple structures such as water wheels, hand mills and other objects operated by humans or animals [38]. The term flywheel was first used around 1784 [39]. At the beginning of the industrial revolution, flywheels were used in boats and trains with steam engines and as energy accumulators in factories [40]. With the development of technology in the 20th century, flywheels have been developed and improved, thanks to which they were used in transport (gyrobuses) in the 1950s [41] and in the following years in electric vehicles, emergency power systems and space missions [42]. In energy storage with flywheels, electric energy is converted into mechanical kinetic energy accumulated in the rotational motion of the rotating mass [43]. The simplified scheme of operation and the basic components of the kinetic energy storage are presented in Figure 3.

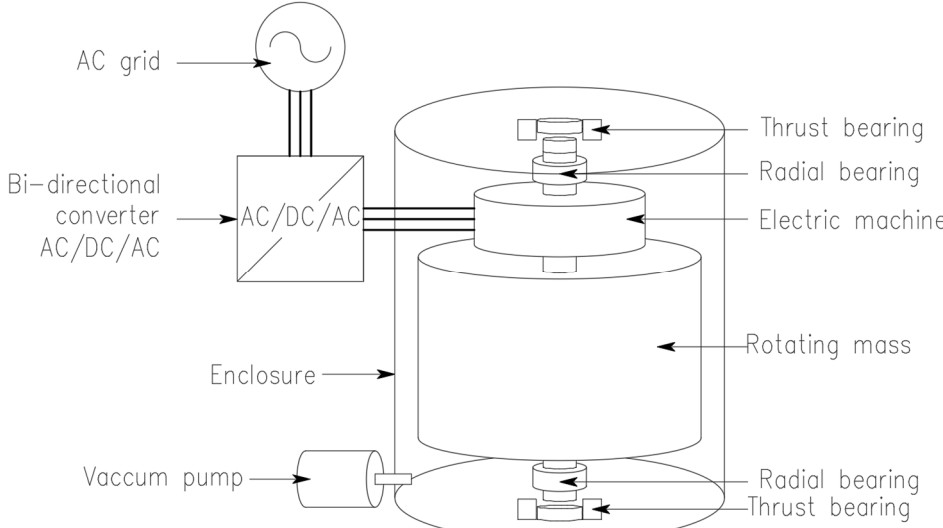

**Figure 3.** Block diagram of kinetic energy storage.

The surplus of electricity is supplied through a bi-directional AC/DC/AC converter to an electric machine operating in motor mode. The motor converts electric energy into mechanical kinetic energy. The flywheel is controlled directly by the electric machine through the coaxial connection of the motor with the rotating mass. While feeding the

energy storage system, the rotating mass is accelerated to the nominal speed of the energy reservoir. In the case of discharge, the electric machine operates in generator mode, enabling the conversion of the accumulated mechanical kinetic energy of the rotating flywheel into electric energy. Discharging the FESS (flywheel energy storage system) causes the flywheel to brake. In order to limit self-discharge, i.e., the process of energy loss during idling, components such as bearings and a vacuum enclosure are used. The energy stored in the kinetic energy store is determined by Formula (3).

$$E_{el} = \eta \frac{1}{2} I \omega^2 \tag{3}$$

where

$\eta$ is total efficiency;
$I$ is the moment of inertia;
$\omega$ is the angular speed.

Currently, energy storage with a flywheel is used in a wide range of power and capacities, from large-scale storage for power grids to small-scale storage for individual customers. The accumulator in the form of a flywheel with a capacity of 150 kWh [44] was used in the Volvo S60 in 2013. The rotating mass made of carbon fiber with a mass of 6 kg and a diameter of 200 mm accelerates to 60,000 rpm, enabling energy recuperation during braking. By using FESS in a car, it is possible to reduce fuel consumption by 25% and obtain an additional 80 KM (approx. 60 kW) [45]. The Beacon Power installation in Stephentown, New York, with a capacity of 5 MWh, is an example of using energy storage in the electric power industry enabling the consumption of 20 kW for 15 min. Launched in 2011, the installation consisting of 200 flywheels is designed to adjust the grid frequency [46]. The increasing popularity of FESS is associated with numerous applications in the area of energy storage and management. Currently, the most important applications of kinetic energy storage are the following:

- Improvement of the quality of electricity [47];
- Frequency adjustment [48];
- Voltage picks control [49];
- Uninterruptible power supplies (UPS) [50];
- Electromobility and transport [51];
- Space industry [52];
- Military industry [53];
- Renewable energy sources [54].

A summary of information on kinetic energy storage is presented in Table 5.

**Table 5.** Advantages and disadvantages of kinetic energy storage [37–46].

| Advantages | Disadvantages |
| --- | --- |
| - Fast response time;<br>- No impact from the external temperature on the operation of the system;<br>- No use of environmentally harmful substances in the construction of FESS;<br>- Long service life of approx. 20 years with proper maintenance;<br>- Short charging time. | - Danger of explosion during failure;<br>- The need for regular maintenance;<br>- The need to provide tensile strength to the rotating mass;<br>- Relatively high internal losses due to friction forces;<br>- Short discharge time;<br>- Limited energy storage time of only approx. 15 min. |

### 3.3. Battery Energy Storage

The development of industries, especially the energy and electromobility industries, has forced the development of cheap and easy-to-use electricity storage [55]. This has resulted in the dynamic development of the battery technology market. The reasons for the growing popularity of this technology are as follows:

- Availability resulting from the existing infrastructure for the production and transport of batteries [15];
- Maturity of the technology, thanks to numerous tests and studies, as well as protective devices enabling safe operation of cells [56];
- Electrochemical energy conversion allowing charging and discharging in the same technology [16];
- Possibility of modular design [57];
- Good current–voltage parameters of cells [56].

Thanks to the above advantages, battery cells are popular in industries as energy storage devices for stationary applications (energy storage and uninterruptible power sources (UPS)) [16] as well as for electromobility and power supply of mobile machines [58]. Currently, various battery technologies that have different properties and applications are commonly used, as presented in Table 6.

**Table 6.** Comparison of battery technologies [16,58–61].

| Group | Type | Advantages | Disadvantages |
|---|---|---|---|
| Ion-lithium | NMC | • High availability on the market;<br>• Market maturity;<br>• High density of energy. | • Technology widely regarded as dangerous;<br>• High price;<br>• Low average number of charging cycles;<br>• Limited availability due to the high demand of the electromobility industry;<br>• Costly recycling. |
| Lithium iron phosphate | LFP | • In emergency situations, there is no explosion or open fire [53];<br>• Approved for use in potentially explosive environments;<br>• Low price. | • The need to protect against deep discharge;<br>• High operating costs with frequent charging and discharging (>1 cycle per day);<br>• Low average cell life;<br>• Degradation;<br>• Costly recycling. |
| Lithium titanate | LTO | • Widely regarded as a safe technology [56];<br>• Long service life compared with other technologies;<br>• High efficiency of cells. | • Relatively high cost of cells;<br>• Limitations in use related to the cost of cells;<br>• Costly recycling. |
| Lead–acid | LAB | • Commonly used in industry;<br>• Low cost of cells;<br>• Mature cell-recycling technologies. | • High operating costs;<br>• Small number of cycles;<br>• Cell degradation at deep discharge;<br>• Limited range of use due to technology flaws. |
| Vanadium redox flow | VRFB | • Low operating costs;<br>• Considered to be safe [54];<br>• Negligible cell degradation;<br>• Practically unlimited number of cycles;<br>• Scalable for power and capacity;<br>• Negligible self-discharge;<br>• 100% recyclable. | • Relatively new technology;<br>• Low energy density;<br>• Low currents compared with capacitance. |
| Sodium ion | SIB | • Good availability of materials for battery production;<br>• Lower battery cost (USD 40–77 per kWh);<br>• Regarded as safe, flame retardant and non-leaking;<br>• There is no risk of uncontrolled temperature rise and ignition;<br>• Possible for using in large-scale energy storage. | • Low electrochemical kinematics of the cells;<br>• Lower energy density (75–200 Wh/kg);<br>• Fewer cycles (up to 5000) compared with lithium batteries;<br>• Lower single cell voltage (2.5 V). |

Currently, battery cells are commonly used in portable electronic devices (laptops, mobile phones and tablets), transport (electric and hybrid cars), special machines for the mining industry and stationary energy storage [57,62]. The dynamically growing demand for battery energy sources in 2018 amounted to 180 GWh; it is expected that by 2030 the demand may even reach 2600 GWh. However, after taking into account the reduction of cell costs, it is estimated that the demand for battery energy storage may increase to 3562 GWh [63]. The popularization of battery energy storage is mainly related to pro-ecological activities, such as the departure from conventional energy sources in favor of renewable energy sources and the promotion of electric cars. The use of different types of batteries due to the features of each technology (for example, energy storage battery containing lithium cells) allows operation in systems requiring high temporary power and quick responses. An example of such a system is the installation located in the German town of Cremzow, with a power of 22 MW and a total capacity of 35 MWh. The battery energy storage is used to control the frequency in the grid (primary control service). The flow-through cell is another type of battery characterized by a long working time, enabling long-term energy storage thanks to the almost non-occurring self-discharge. An example of such an installation is a flow energy storage in the city of Jeonju in South Korea, with a power of 260 kW and a capacity of 2.2 MWh. The system is part of the installation for powering the paper mill and is intended for arbitrage, i.e., charging the batteries in an economically favorable period for the purchase of electricity and discharge at higher electric energy prices. The energy storage reservoir is charged during the economical night-time electricity tariff and unloaded during the day when the factory is working [16].

### 3.4. Hydrogen Energy Storage

Hydrogen energy storage is a newly developed energy storage technology. Energy accumulation takes place in three stages: hydrogen production using electrolyzers, hydrogen storage in special pressure tanks or in the form of chemical compounds and hydrogen conversion into electricity via fuel cells. The efficiency of such a process is currently estimated at 25–45% [15]. A simplified scheme of the operation of hydrogen energy storage devices is presented in Figure 4 [64].

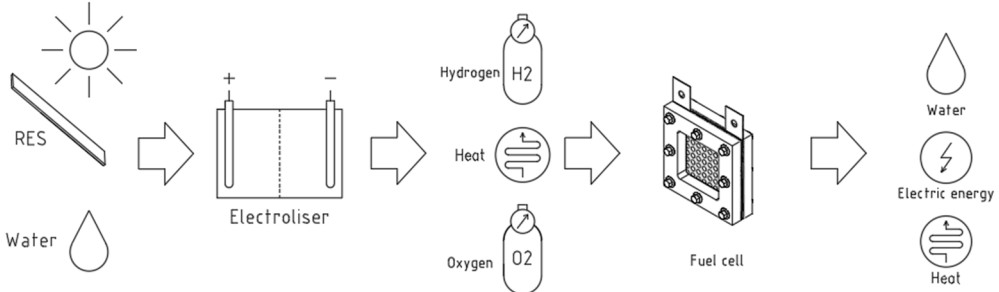

**Figure 4.** Diagram of energy storage in hydrogen reservoirs.

Surplus energy from renewable energy sources (RES) is used in water electrolysis to produce hydrogen. Additional products are heat and oxygen, which can also be stored in tanks or emitted to the atmosphere. During the demand for electricity, hydrogen and oxygen are fed to the fuel cell, where the process of reverse electrolysis of water takes place to produce electricity. Additional reaction products are water and heat. Battery energy storage is often used in conjunction with hydrogen energy storage to protect the system against sudden and temporary load changes [65]. The capacity of the energy storage is related to the hydrogen storage capacity in the tanks. Losses in energy conversion are reduced by using the generated thermal energy for utility purposes. Installations enabling the storage of electric energy through hydrogen and the use of waste heat for utility purposes achieve an efficiency of 90% [66]. Hydrogen energy storage reservoirs enable long-term energy storage and low costs of storage units' scalability, so it is possible to use

them in off-grid solutions, i.e., energy-independent from the external grid. In the case of hydrogen energy storage, scalability is possible.

Currently, solutions for individual customers and enterprises are available. The Home-powersolutions (HPS) company, which offers the PICEA solution, offers examples of solutions designed for individual customers. The manufacturer offers hydrogen energy storage solutions consisting of a module intended for use inside the facility, operating as an electrolyzer and a fuel cell, as well as a management system and a hydrogen storage unit intended for use outside the facility for safety reasons.

The energy storage reservoir has a basic capacity of a hydrogen energy storage reservoir of 300 kWh, which can be extended to 1500 kWh, and a battery storage system with a capacity of 20 kWh as a short-term energy storage. Nominal power of the battery storage system is 7.2 kW, while the continuous power of the fuel cell is 1.5 kW. The tank also enables using the thermal energy in the range of 2000–4000 kWh per year. The energy storage unit enables working with photovoltaic installations and a $3 \times 400$ V grid. According to the data provided by the manufacturer of the solution, the estimated investment cost is EUR 45,000–79,000 [67].

Other solutions for hydrogen energy storage are the solutions from GKN Hydrogen. The solutions are intended for industrial applications, as seasonal energy reservoirs, for securing energy supplies and for vehicle charging stations. The company offers storage reservoirs in a container designed in three main sizes: HY2MINI, with a capacity of 165–420 kWh, a nominal power of 8 kW and a temporary power of 14 kW (30 min) built in a 10ft container; HY2MEDI, with a capacity of 1000–2000 kWh, nominal power in in the range of 8–24 kW and instantaneous power of 75 kW (30 min) built in a 20 ft container; HY2MEGA, with a capacity of >8600 kWh in the form of a hydrogen tank enabling the accumulation of 260 kg $H_2$. The HY2MEGA solution uses external fuel cells and electrolyzers. As in the case of HPS solutions, GKN Hydrogen storage tanks combine in their design a hydrogen energy storage unit with a battery energy storage. The output voltage of HY2MINI and HY2MEDI is 120/230/400 V 50 Hz [68]. The strengths and weaknesses of this energy storage technology are shown in Table 7.

**Table 7.** Advantages and disadvantages of hydrogen energy storage [63–68].

| Advantages | Disadvantages |
|---|---|
| <ul><li>Virtually zero-emission hydrogen production and processing;</li><li>Use of water for the hydrogen production;</li><li>Scalability of the solution by increasing the capacity of hydrogen tanks;</li><li>Possibility of long-term energy storage;</li><li>High energy density;</li><li>Possibility of using waste heat for utility purposes.</li></ul> | <ul><li>Low efficiency of 25–45%;</li><li>High investment costs regarding the hydrogen production and processing equipment (fuel cells and electrolyzers);</li><li>Danger of gas ignition or explosion in the case of a leak in the system;</li><li>Low energy density in volume, need for using large tanks;</li><li>Need to use compressors and high-pressure tanks dedicated to hydrogen storage.</li></ul> |

### 3.5. Supercapacitors

The supercapacitor is a technology of energy storage using electrolytic capacitors of a special design. Energy storage reservoirs using this technology are characterized by a low energy density of 10 Wh/kg with one of the highest energy storage efficiencies of 95% [69]. A durability of 20 years and a large number of charging discharges (up to 1,000,000 cycles) are additional advantages of supercapacitors in energy storage applications [70]. Supercapacitor storage is commonly regarded as short-term storage, as it allows for quick charging and discharging. Currently, supercapacitors are most often used in combination with other energy storage devices, e.g., batteries, fuel cells. The advantage of such a connection is the possibility of a short-term supply of peak power, which stabilizes the operation of other energy storage devices, reducing the size of the energy storage reservoirs and increasing

the life of the combined energy storage system. One of the most popular hybridizations is the combination of supercapacitors with battery cells, which takes place in the following topologies [71]:

Passive (P-HEST);
Discrete (D-HEST);
Active (A-HEST).

Supercapacitors, which were initially used only to support the power supply of memory systems, are now gaining more and more new areas of application. The basic industries using this technology include power industry (stabilization of grid parameters), telecommunication (power for telephones), automotive industry (power components for electric and hybrid cars) and space industry (energy storage) [71].

An example of a solution used in the industry is a pack of supercapacitors with the symbol XLR-51 made by EATON, with dimensions of $177 \times 421 \times 196$ mm and a weight of 14.7 kg. The storage reservoir is equipped with two electric connectors of the main current path in the form of M10 terminals. A sample pack of supercapacitors with a nominal DC voltage of 51.3 V and an energy capacity of 188 F enables the maximum pack current of 2485 A, i.e., 131.6 kW of instantaneous power. The presented storage reservoir allows up to 1,000,000 charge and discharge cycles. The capacity of the tank is determined according to Formula (4) at 68.7 Wh [72].

$$E_{el} = \frac{0.5 \, C \, V^2}{3600} \tag{4}$$

where $E_{el}$ is the energy accumulated in the reservoir (Wh), $C$ is the electric capacity in (F) and $V$ is the maximum operational voltage in ($V$).

A characteristic feature of supercapacitors is the possibility of a high current flow at a small capacity of the energy storage reservoir. A large number of work cycles and the possibility of deep discharges enable the use of supercapacitors as short-term energy storage in hybrid energy storage systems (Table 8).

**Table 8.** Advantages and disadvantages of *supercapacitors* [69–72].

| Advantages | Disadvantages |
|---|---|
| <ul><li>Fast response time;</li><li>High number of charge and discharge cycles of more than 500,000;</li><li>No use of harmful substances in the construction of supercapacitors;</li><li>High efficiency during charging and discharging 85–98%;</li><li>High output power due to low internal resistance;</li><li>Wide operating temperature range.</li></ul> | <ul><li>Low energy density 3–5 kWh/kg;</li><li>Low single supercapacitor voltage of approx. 2.7 V;</li><li>The need to use advanced equipment to control and manage supercapacitor packages;</li><li>High self-discharge compared with electrochemical batteries;</li><li>Linear supercapacitor discharge voltage characteristics;</li><li>Higher price compared with lithium batteries of similar size.</li></ul> |

## 4. Hybrid Systems of Energy Storage

Dynamic changes in power systems, popularization of renewable energy sources and the need to ensure uninterrupted supplies of electrical energy make it necessary to use energy storage devices of differentiated properties. In addition, the investment related to the acquisition of the warehouse must be profitable and fully functional in order to increase the comfort of each consumer. To ensure full functionality, it is necessary to combine various technologies in such a way to ensure adequate versatility. Hybrid energy storage systems (HESS), combining two or more energy reservoir technologies of complementary properties such as fast response time, negligible self-discharge and a large number of work cycles, may be the solution [16].

*4.1. Hybrid Energy Storage within the HyStore Project*

Currently, research work on hybrid energy storage is being conducted by a team of experts from the Institute of Fluid-Flow Machinery of the Polish Academy of Sciences. In 2018, a hybrid energy storage test stand was built at the KEZO research center in Jabłonna (near Warsaw, Poland) [73]. HyStore is an energy management system designed for use in energy storage, especially solutions combining various energy storage technologies, thus creating a hybrid energy storage.

The total capacity of the energy storage units included in the HyStore system is 180 kWh, while the maximum power is 60 kW. The hybrid storage system integrates four energy storage technologies [74]:

- Flow battery with a power of 12 kW and a capacity of 100 kWh;
- LiFePO$_4$ lithium iron phosphate battery with 24 kW and a capacity of 24 kWh;
- Carbon–lead battery with a power of 12 kW and a capacity of 24 kWh;
- Lead–acid battery with a power of 12 kW and a capacity of 32 kWh.

The hybrid energy storage system is part of the microgrid of the research center and is used to balance energy obtained from renewable energy sources. The microgrid operating in the KEZO research center enables to cover the demand for electricity thanks to the use of various renewable energy technologies, e.g., PV installations with a capacity of 180 kW, wind turbines with a capacity of 12 kW and a cogeneration source (CHP) with a capacity of 100 kW. The microgrid provides power to recipients of various types of energy consumption, e.g., office and laboratory buildings, HVAC equipment and electric vehicle chargers. The microgrid ensures the correct flow of energy between the mentioned RES sources, loads and Hystore hybrid energy storage. The presented system is a facility that is tested to select the hybrid energy storage configuration for the needs of the customers while maintaining the long life of the batteries. The hybrid energy storage operating at the KEZO Research Center of the Polish Academy of Sciences in Jabłonna is part of the HyStore project. HyStore is an energy management system (EMS) designed for energy storage in various technologies integrated with each other.

The energy flow is managed by a local controller and a network application (optimization server). The local controller is responsible for direct battery control, energy balancing, improvement of network parameters' quality, battery operation management and ensuring continuity of power supply in the case of a failure (island operation). The role of the optimization server, on the other hand, is to determine the energy storage operation schedule, current financial analyses (price arbitrage) and prediction of the energy balance, including energy production from RES and current energy consumption [68]. The block diagram of the HyStore system is shown in Figure 5.

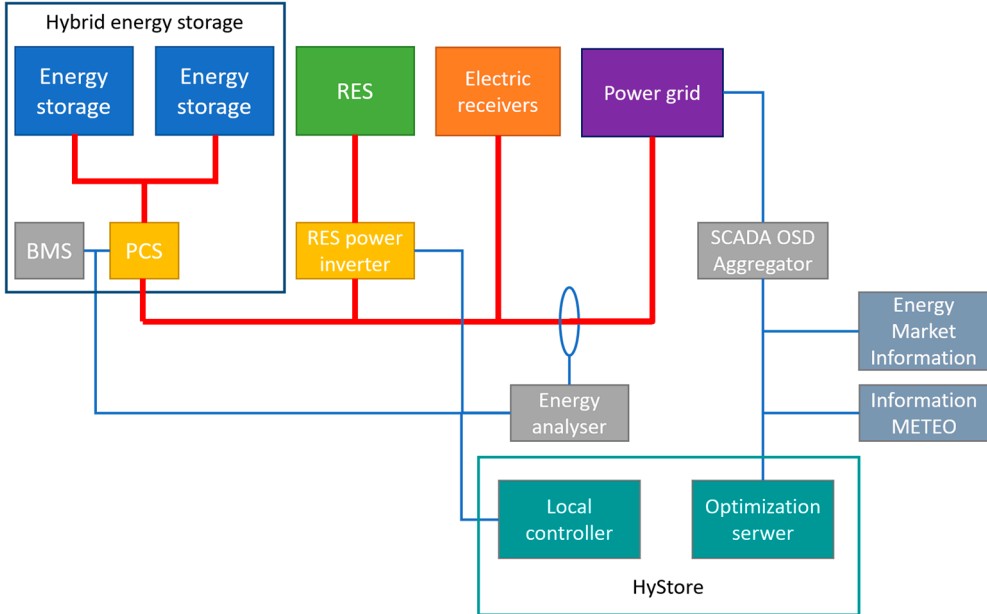

**Figure 5.** Block diagram of the HyStore system.

*4.2. A Hybrid Energy Storage System Integrated with a Railway Conditioner*

The system was developed to reduce the risk occurring from the operation of high-speed railways, causing a series of impulses negatively affecting the parameters of the power grid and additionally increasing the operational costs of traction substations [18]. The direct reason for the appearance of impulses in the network is the nature of high-speed rail operation. High-speed rail tractions are predominantly powered from single-phase grids, which contribute to negative effects such as increasing generator losses, protection failures and transformer losses. In order to reduce the negative effects of high-speed railways on the distribution network, the HESS-RPC system was suggested, consisting of a hybrid energy storage including a magnetic energy storage (SMES), a battery energy storage and a railway power supply conditioner (RCP)17. Figure 6 shows a block diagram of the HESS-RCP system.

The HESS-RCP system [75] shown in Figure 6 is powered from a 220 kV high voltage electrical connection by a three-phase traction transformer with a KT ratio. The high voltage is reduced to two phase voltage sources of 27.5 kV powering both transmitters. The railway power supply conditioner (RCP) consists of two single-phase back-to-back voltage converters connected to each other via a DC bus with an attached capacitor, powered by two transformers reducing the voltage of 27.5 kV line. The hybrid energy storage system consists of a battery energy storage and a magnetic energy storage (SMES) [18]. The HESS is connected to the T&A by a DC bus through an appropriate interface circuit. The use of energy storage in the form of HESS enables minimization of peak power by managing the energy flow in such a way to reduce the effects of impulses in high-speed railway installations. By configuring the system, it is possible to determine the maximum value of the load power (Pmax) and work in the following three modes:

- Charging when P > Pmax;
- Standby when P = Pmax;
- Discharging when P < Pmax.

The use of HESS-RCP in high-speed railway power supply systems will enable the following:

- Minimization of the peak power of the traction substation, while the power of the traction transformer and the connection can be reduced, which is associated with a reduction in the operating costs of the traction substation.
- Improvement of the quality of network parameters: compensation of reactive power and reduction of disturbances transmitted to the network.

- Hybridization of energy reservoirs will improve the service life of the battery pack by lowering the power and reducing the number of charging and discharging cycles.

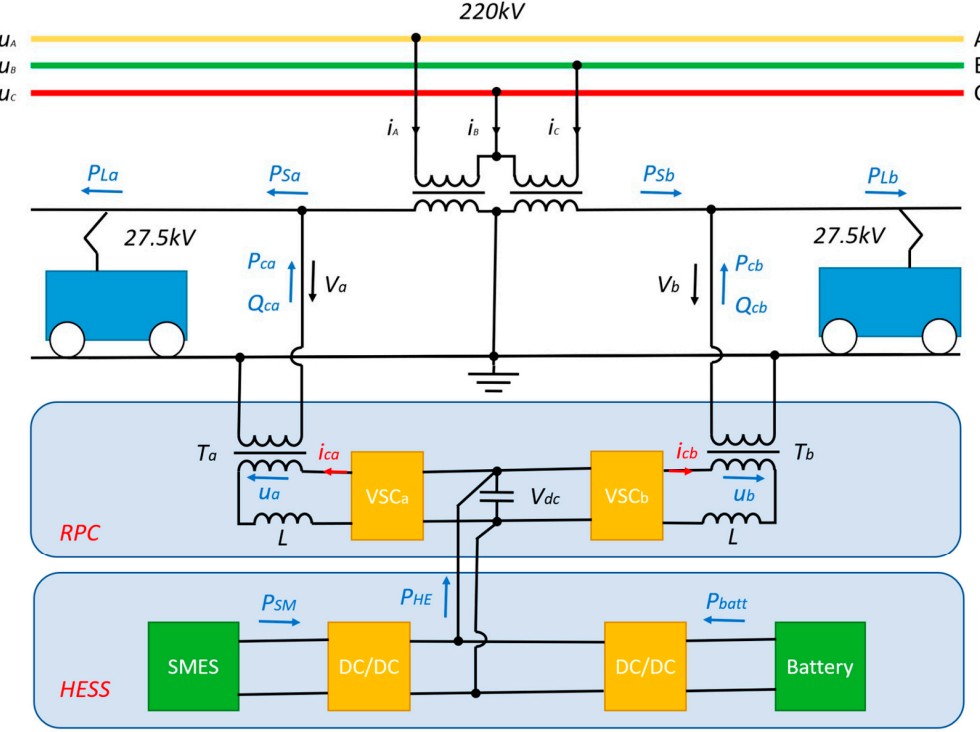

**Figure 6.** Block diagram of HESS-RCP system [72].

### 4.3. Intelligent Hybrid Energy Storage with Power Electronic Converters Using High-Voltage Low-Inductance SiC Mosfet Power Modules

Hybrid energy storage presented by MARKEL Sp. z o. o. is based on the use of a power electronic unit (EBM). The EBM system manages the energy flow between RES, energy storage units, energy consumers and the distribution grid. The system consists of two basic modules: U1 and U2. The U1 module is responsible for managing the energy flow between each energy storage system. The U2 module is responsible for connecting to the distribution grid and AC receivers [76]. The scheme of the hybrid energy storage with EBM is shown in Figure 7.

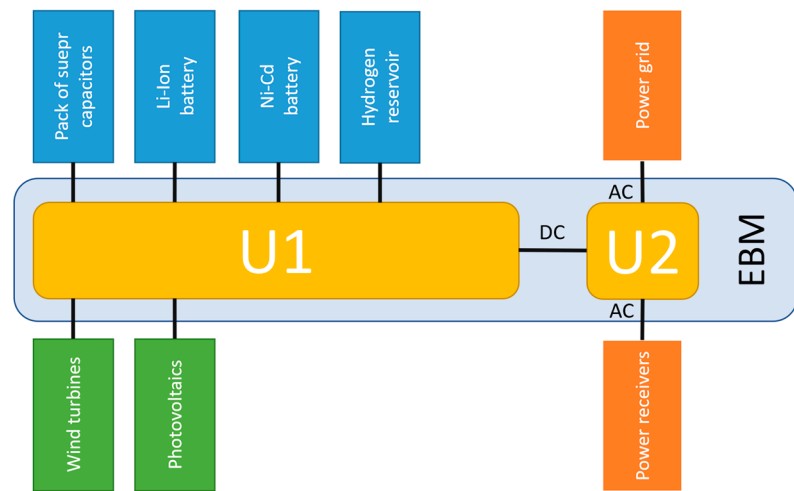

**Figure 7.** Hybrid energy storage from RES [76].

In order to ensure high dynamics of energy flow management with a simultaneous reduction of EMC disturbances, it is necessary that the design of the EBM module be integrated inside one enclosure of the smallest possible dimensions that is economically viable with high efficiency. To implement the presented assumptions, the EBM uses low-inductance SiC MOSFET power transistors that enable switching with a frequency in the range of 10–50 kHz at a voltage of up to 4 kV [77]. The EBM system enables control of the energy flow in the U1 module using the DC bus, which additionally enables the connection of DC receivers without loading the distribution grid [78] and, if required, galvanic isolation [79]. The U1 module is connected to the AC installation (distribution grid and AC receivers) via a bi-directional AC/DC converter. A transformer is used in the U2 module to ensure galvanic separation between the U1 module and the alternating current installation. The operation of the transformer at high frequencies (in the range of 10–50 kHz) makes it possible to reduce the size of the transformer and the cost of manufacturing such a transformer. However, in the case of DC/DC DUAL ACTIVE BRIDGE (DAB) converters, the transformer also acts as a galvanic separator between the DC source or receiver and the common DC bus. The transformer does not directly stabilize voltage.

In the U2 unit, a transformer connected to a power electronic converter is used. The use of the so-called intelligent transformer allows to reduce the dimensions and weight of the transformer operating at an increased frequency much higher than 50 Hz. The dimensions of the transformer, together with power electronic converters, are 3.5 times smaller than the solution with a transformer operating at a frequency of 50 Hz [80]. There are two different methods for controlling the EBM module, depending on the requirements of the recipient of the solution. In simple applications, where the system performs a single function, e.g., maximizing the efficiency of the PV installation, the microprocessor system, that is part of the system, is responsible for controlling the energy flow. For this purpose, the software of the system has been modified to be able to perform the desired single function. In the case of more complex applications, e.g., microgrids, EBM is controlled by a master controller (the SCADA system or an industrial computer). Additional advantages of using intelligent transformers include voltage stabilization, reactive power compensation and connecting installations with different operating frequencies. It is estimated that, with the popularization of the so-called Smart Grid, the number of solutions with smart transformers will increase [81]. An additional element of the intelligent energy storage should be software based on self-learning algorithms that increase the flexibility of adaptation to the current grid load and changes in current tariff, thus increasing the efficiency and profitability of the system.

### 4.4. Multiport DC Power Router

In order to integrate renewable energy sources with battery and hydrogen storage tanks, a multiport DC power router solution is proposed. The proposed power router consists of the following six ports [82]:

- Network port: includes an AC/DC converter interface to connect the mains voltage to the DC bus of the DC router. The port is responsible for the exchange of energy between the energy generated from RES and the AC installation, as well as for stabilizing the voltage on the DC bus. During off-grid operation, the port is disabled.
- Wind energy port: includes an AC/DC converter, the primary side of which is connected to the wind turbine and the secondary side to the DC bus. The port is responsible for receiving energy generated by the wind turbine and converting it to DC voltage with parameters adapted to the common DC bus voltage.
- PV port: includes a DC/DC converter whose primary side is connected to the photovoltaic system and the secondary side is connected to the DC bus of the router. The port is responsible for adjusting the voltage of the PV installation to the common voltage of the system.
- Battery energy storage port (ESS Port): includes a DC/DC converter with the primary side connected to the battery energy storage and the secondary side connected to

the common DC bus. The port is responsible for stabilizing the voltage on the DC bus in the case of a power failure from the grid port, i.e., working in off-grid mode. The battery energy storage also ensures reliable operation of the router in the case of a power failure.

- Hydrogen port: includes DC/DC converters, the primary side of which is powered from the DC router and the secondary side is connected to the electrolyzer. The port is responsible for controlling hydrogen production by controlling the current and power of the electrolyzer. It is possible to use different types of electrolyzers, including the most popular PEM and alkaline ones.

The block diagram of a six-port energy router designed for hydrogen production is shown in Figure 8.

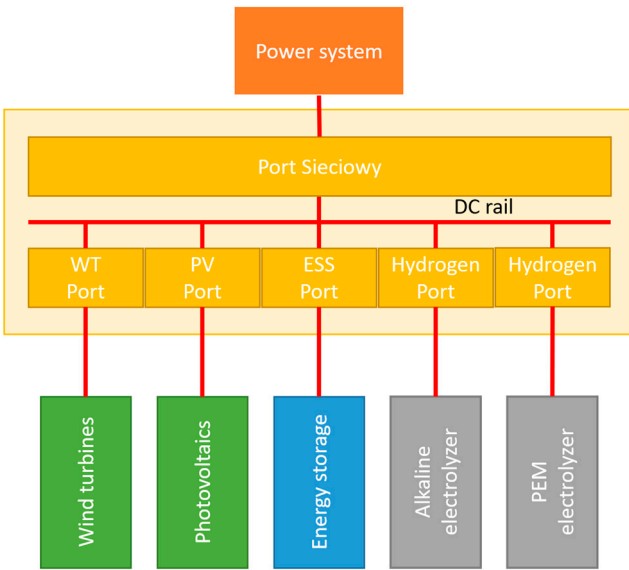

**Figure 8.** Block diagram of DC router [82].

Use of a DC router in hydrogen production enables stabilizing the operation of electrolyzers that require constant voltage conditions for proper operation, high efficiency and life. An additional advantage of using the DC router is a possibility of the maximum use of energy from RES (PV panel or wind turbines) [82]. The security aspects of hydrogen storage are not the subject of the presented concept of a multiport DC router; however, it is recommended to use specialized hydrogen storage tanks and explosion-proof equipment (EX IIC).

### 4.5. Fast Charging Station for Electric Vehicles

Hybridization of energy reservoirs is used in the patent for fast charging stations for electric vehicles intended for use in public places, e.g., car parks, bus depots or gas stations. The charging station being the subject of the invention consists of a hybrid energy storage, a mains power supply system equipped with an AC/DC converter and a power supply system from renewable energy sources (a photovoltaic installation with a DC/DC chopper and a wind turbine integrated with an AC/DC rectifier). The integration of components intended for generation, storage and receiving the energy takes place via a DC bus. The energy storage is equipped with two outputs: a DC output and an AC output through a three-phase inverter. A microprocessor system controls the energy flow through each component of the system. The hybrid energy storage used in this invention consists of a pack of accumulators connected in parallel with a pack of supercapacitors. The hybrid energy storage unit, together with the power electronics, is installed in a cabinet located inside the shelter. The three-phase AC connection and the DC connection are led outside the enclosure. The block diagram of the installation intended for charging the electric vehicles is presented in Figure 9.

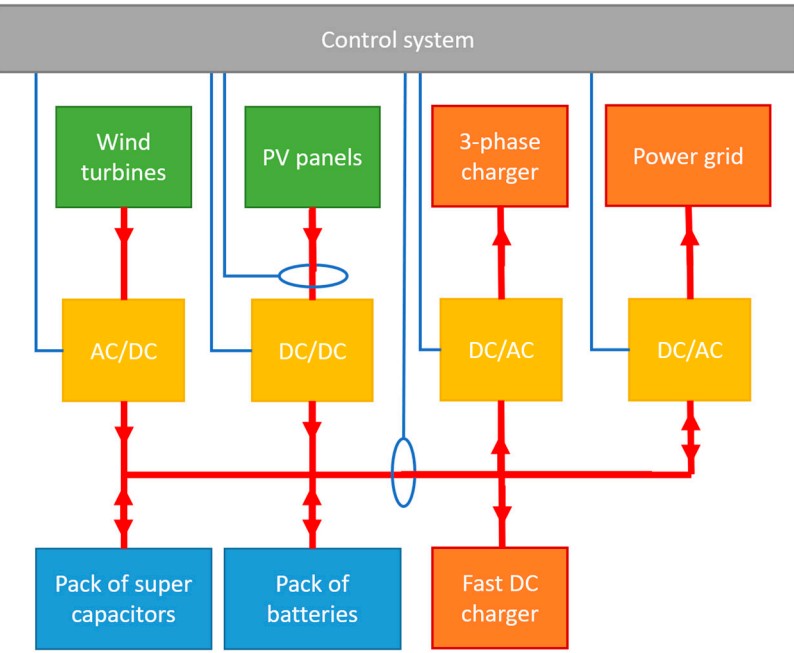

**Figure 9.** Block diagram of charging station [83].

A shelter with installed renewable energy sources—two wind turbines and a PV installation—is the suggested solution for the vehicle charging station. The selection of RES parameters installed in fast vehicle charging stations is closely related to the target location (the possibility of using wind turbines) and the size of the target station (available roof surface area for placing the PV installations). The control system, power electronics and energy storage are placed in a cabinet inside the shelter. Due to high efficiency and no need for excitation, it is advantageous to use generators with permanent magnets cooperating with a rectifier connected to a DC bus. Photovoltaic panels are integrated with the system through a DC/DC chopper. In addition, the system can be supplied from the mains through a bi-directional AC/DC converter. A hybrid energy reservoir consisting of batteries and supercapacitors is connected directly to the DC bus.

The system enables charging the electric vehicles directly with DC voltage and through a unidirectional AC/DC converter with alternating current. Vehicles are charged with high power using the energy stored in the energy reservoir. The hybrid energy storage system is charged mainly from RES, i.e., installed wind turbines and photovoltaic panels. In addition, the reservoir can be charged from a grid with a limited connection power of up to 20 kW. The parameters of the fast charging stations for electric vehicles are selected individually for the recipient's requirements, power connection conditions and the availability of energy from RES. It is therefore impossible to unambiguously determine the universal parameters of energy storage for such an application. The presented solution of the vehicle charging station enables a shortened time of car charging resulting from the use of high power. The energy stored in the reservoir mainly comes from zero-emission renewable sources, with the possibility of charging from the power grid.

## 5. Discussion

The dynamic development of the RES sector in the Polish power system is necessary due to the gradual departure from conventional coal or gas-fired power plants. The use of large RES installations is a great challenge for the power system due to the limited control possibilities and the unpredictable and periodic nature of the operation of renewable energy sources. Energy storage in various technologies can be a tool to reduce the undesirable characteristics of RES. Energy storage systems connected to RES installations will enable the maximum use of the produced power, reduce the risk of destabilization of the power system and provide the owner of the installation with partial or complete

energy independence. By combining various energy storage technologies, it is possible to adjust the operation of the installation to the specific requirements of the consumer, while maintaining the service life of each energy storage device. With the further development of power industry based on renewable sources, it is necessary to develop and implement state-of-the-art installations equipped with hybrid energy storage systems adapted to the nature of production and consumption.

## 6. Conclusions

The dynamic development of the renewable energy sector in Poland, together with the gradual decarbonization of the power industry, makes it necessary to develop and implement state-of-the-art power supply systems. The reasons for this are the risks associated with the increasing number of RES installations using solar and wind energy, i.e., the need to increase the capacity of transmission lines and transmission equipment (e.g., transformers) to collect surplus energy from RES. Another reason is the improvement of the quality of grid parameters, such as the voltage, which is influenced by PV inverters that return energy to the grid. Modern buildings should be equipped with RES microinstallations, e.g., photovoltaics, energy storage and an energy management system to increase the energy independence of the building and energy balancing within the building microgrid. Currently, the highest components of the electricity price are the costs associated with its production and transmission, which is why it is advisable to resign from central energy storage in favor of small local energy storage powered by RES. Thanks to the use of individually configured hybrid energy storage systems, it is possible to adjust them to the specific requirements of the recipient, e.g., complete energy independence of the building, maximum use of energy from RES or price arbitrage. This configuration of the power grid also minimizes the risk of power failure when one of the components fails, because, in the case of power failure, the system enables island operation, thus ensuring uninterrupted access to electricity.

**Funding:** This research received no external funding.

**Data Availability Statement:** No new data were created or analyzed in this study. Data sharing is not applicable to this article.

**Conflicts of Interest:** The authors declare no conflict of interest.

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
