# Peer review of "Overview of Hybrid Energy Storage Systems Combined with RES in Poland"

_energies, doi:10.3390/en16155792_

Round 1
Reviewer 1 Report
The paper under review is devoted for overview of hybrid energy storage systems combined with renewable energy sources. It classifies the energy storage technologies into pumped-storage power plants, kinetic energy storage, battery energy storage, and hydrogen energy storage.
The different hybrid systems of energy storage were also reviewed including hybrid energy storage within Hystore project, hybrid energy storage system integrated with a railway conditioner, intelligent hybrid energy storage with power electronic converters using high-voltage low-inductance SiC Mosfet power modules, multi-portowy router pradu statego, and fast charging station for electric vehicles.
For completeness, the authors are requested to respond to the following queries in the revised version of the paper:
1- In hybrid energy storage within Hystore project:
Does the project name refer to the large storage system in the project?
What is the power capacity of the project? What are the loads served by the project?
The hybrid energy storage system is a part of a microgrid. What are the details of this microgrid?
2- In hybrid energy storage system integrated with a railway conditioner:
How does the use of HESS-RCP system enable minimization of the peak power of the traction substation?
It is stated that the high speed railways causes a series of impulses negatively affecting the parameters of the power grid. What is the origin of these impulses?
3- In intelligent hybrid energy storage with power electronic converters using high-voltage low-inductance SiC Mosfet power modules:
How does the EBM system manage the energy flow between RES, energy storage units, energy consumers and the distribution grid?
The module U2 is responsible for connecting the distribution grid to the AC receivers. In the U2 module, reference was made to an intelligent transformer operating at high frequency in the range 10-50 kHz. How does this transformer operate to do voltage stabilization?
4- In multi-portowy router pradu statego:
What does pradu statego refer to?
What are the precautions made to store hydrogen in the tanks?
5- In fast charging station for electric vehicles:
The hybrid energy storage unit consists of a pack of accumulators and a pack of supercapacitors, W hat is the rating of each accumulator and supercapacitor? What is the number of accumulators and supercapacitors in the pack?
What is the rating of the wind turbine and the PV installation?
Reviewer 2 Report
This manuscript summarizes a comprehensive overview of various energy storage technologies and introduces the concept of hybrid energy storage systems. It provides insightful examples of installed applications using different energy storage devices, effectively highlighting the advantages and providing application recommendations for each technology. While the manuscript already covers a wide range of topics, some additional details are recommended for inclusion before publication.
1. To further enrich the discussion, please provide more specific examples or case studies that can support the information provided. These examples will provide practical illustrations of the concepts discussed.
2. To enhance the reader's understanding, please add technical details and specific features of each energy storage technology. This additional information will help readers grasp the intricacies and capabilities of each technology more effectively.
3. It is suggested to modify the introduction to clearly state the goals and objectives of the article, which will help readers understand the purpose and direction of the content from the outset.
4. To improve the summary and conclusions, it is suggested to add key findings or major takeaways. This will provide a concise and memorable summary of the main points discussed throughout the article.
5. It would be valuable to incorporate specific problems and solution paths related to future renewable energy applications. This will demonstrate the practicality and relevance of the discussed energy storage technologies in addressing real-world challenges.
6. Please add a comparative summary of the advantages and disadvantages of different energy storage technologies. This will provide readers with a clear understanding of the pros and cons of each technology.
7. Some related literature may be helpful for the manuscript, such as Nat. Commun., 2023, 14, 3701; Adv. Mater. 2023, 35, 2203547; Adv. Energy Mater. 2023, 13, 2300648.
Moderate editing of English language required.
